# Functional Changes of White Matter Are Related to Human Pain Sensitivity during Sustained Nociception

**DOI:** 10.3390/bioengineering10080988

**Published:** 2023-08-21

**Authors:** Hui He, Lan Hu, Saiying Tan, Yingjie Tang, Mingjun Duan, Dezhong Yao, Guocheng Zhao, Cheng Luo

**Affiliations:** 1The Clinical Hospital of Chengdu Brain Science Institute, MOE Key Lab for Neuroinformation, University of Electronic Science and Technology of China, Chengdu 610054, China; hehui_hhwdwd@163.com (H.H.); diuzhuren_hl@126.com (L.H.); tsy1448574502@163.com (S.T.); hahajiege1@163.com (Y.T.); fourhospital@163.com (M.D.); dyao@uestc.edu.cn (D.Y.); 2Research Unit of NeuroInformation, Chinese Academy of Medical Sciences, Chengdu 610041, China; 3High-Field Magnetic Resonance Brain Imaging Key Laboratory of Sichuan Province, School of Life Science and Technology, University of Electronic Science and Technology of China, Chengdu 610056, China

**Keywords:** pain sensitivity, brain white matter, functional connectivity, anxiety

## Abstract

Pain is considered an unpleasant perceptual experience associated with actual or potential somatic and visceral harm. Human subjects have different sensitivity to painful stimulation, which may be related to different painful response pattern. Excellent studies using functional magnetic resonance imaging (fMRI) have found the effect of the functional organization of white matter (WM) on the descending pain modulatory system, which suggests that WM function is feasible during pain modulation. In this study, 26 pain sensitive (PS) subjects and 27 pain insensitive (PIS) subjects were recruited based on cold pressor test. Then, all subjects underwent the cold bottle test (CBT) in normal (26 degrees temperature stimulating) and cold (8 degrees temperature stimulating) conditions during fMRI scan, respectively. WM functional networks were obtained using K-means clustering, and the functional connectivity (FC) was assessed among WM networks, as well as gray matter (GM)–WM networks. Through repeated measures ANOVA, decreased FC was observed between the GM–cerebellum network and the WM–superior temporal network, as well as the WM–sensorimotor network in the PS group under the cold condition, while this difference was not found in PIS group. Importantly, the changed FC was positively correlated with the state and trait anxiety scores, respectively. This study highlighted that the WM functional network might play an integral part in pain processing, and an altered FC may be related to the descending pain modulatory system.

## 1. Introduction

Pain is a complex physiological and psychological feeling involving sensory-discriminative, cognitive-evaluative, affective-motivational and motor networks in the human body [1]. Several mental disorders, such as anxiety, depression and sleep disorders, are induced by long-term painful perception. The emotional state also has an enormous influence on pain: a negative emotion increases painful perception, whereas positive emotion reduces pain [2]. Many studies found that placebo analgesia was associated with functional activity within painful-sensitive brain regions [3,4]. These findings provide evident that placebos can alter the experience of pain. Moreover, Wiech and colleagues indicated that the brain network could regulate attention, expectation and reappraisal related to painful cognition through the descending pain modulatory system [5]. In addition, functional tissue of white matter (WM) is also associated with pain processing [6]. This suggests that the WM functional network is feasibly involved during the modulation of pain.

The effects of cognition on the descending pain modulatory system have been examined extensively in neuroimaging studies. The periaqueductal grey (PAG) plays a key role in the modulation of pain processing [7]. It receives direct (and indirect) nociceptive input from the dorsal horn and sends projections to the inferior olive as well. Based on these connections, PAG will deliver sensory and motor inputs to the cerebellum [8,9]. Some researchers documented that pain could be modulated by the cerebellum and found that chemical stimulation applied to the cerebellar fastigial nucleus decreases visceral nociceptive reflexes [10]. The pain’s unpleasantness was positively correlated with an increased activation of the posterior cerebellum, suggesting that the posterior cerebellum may be a key region in the cognitive assessment of pain [11]. In summary, the cognitive region in the cerebellum may be related to the encoding of pain, possibly as a cognitive modulator. In addition, Qin and colleague found that an interruption of a connection during pain processing may be associated with abnormal function of the WM bundle [12]. However, the relationship between the WM functional network and the top-down pain modulatory network is unclear.

Previous studies mainly used diffusion tensor imaging (DTI) to study the relationship between the WM structure and pain. The function of WM was rarely assessed during the painful stimulation. In patients with chronic musculoskeletal pain, changes of axial diffusivity (AD) in WM were directly related to its symptom severity [13]. The study of patients with acute thermal pain has been found that a decreased fractional anisotropy (FA) of the WM region can affect the functional response to pain [14]. The WM integrity within and between regions of the descending pain modulatory network is critically linked with the individual ability for endogenous pain control [15]. Therefore, WM changes may play a crucial role during pain processing. In a study of orthodontic pain patients, researchers found that WM network could mediate emotional and cognitive networks during pain stimulation [6]. This finding provides a basis for studying the changes of the WM functional network in the process of pain. Although there is little research on the WM function network of pain, its significance may be great and important.

There are mainly two different experimental pain models. One model uses electrical stimulation to induce pain perception [16]. Another model induces pain through cold stimuli (CPT: cold pressor test) [17,18]. Cold pressor pain has been shown to be opioid sensitive [19]. Moreover, the descending pain modulatory system is implicated in opioid analgesia [3]. Thus, in this study, subjects were given cold pain stimulation to induce the transmission of pain information by the descending pain modulatory system. We hypothesized that the WM functional network played a conduction role in the painful regulatory system. Moreover, we estimated the FC within WM functional networks and their relationship to the known GM functional networks. By comparing the differences in WM functional networks between pain-sensitive and pain-insensitive individuals, this study linked WM’s specifically functional characteristic with the intrinsic pain modulatory system in the human brain.

## 2. Study Design

The Strengthening Observational Studies in Epidemiology (STROBE) guideline was followed [20]. Moreover, the study was conducted in accordance with the Declaration of Helsinki, and the protocol was approved by the Ethics Committee of UESTC. To ensure the completeness and accuracy of observational data, a professional master performed experimental procedure. A professional magnetic resonance imaging (MRI) scanner performed data scanning.

### 2.1. Participants

All subjects involved in this study were recruited from January 2021 to February 2021 at the University of Electronic Science and Technology of China (UESTC). All subjects were informed of the experimental procedure and signed the informed consent before the experiment. Experimenter selected the subjects based on the following criteria: male, no smoking history, no acute or chronic pain of any kind, no related cognitive disorders, no metal implants in the body, no other magnetic resonance contraindications such as claustrophobia.

Then, the trait anxiety and state anxiety of all subjects were evaluated through State–Trait anxiety inventory (STAI) [21]. The Chinese version of the STAI has been validated for measuring the trait and state anxiety of Chinese subjects [22]. We ultimately chose to use the Chinese version of the STAI.

### 2.2. Experimental Procedure

The CPT was produced by immersing subject’s right hand and wrist in a 4-L tank of water and crushed ice. The water temperature ranged from 0.5–1.5 °C. During the subject’s experienced cold pressor trial, they were requested to place their right hand and wrist in ice water and to leave it there until the pain reached an intolerable level. Subject was not permitted to remain in the ice water for more than 5 min. The time was recorded. Subjects were free to withdraw from this study at any time. Then, the painful sensitive and insensitive subjects were grouped based on individuals’ CPT tolerance time as follows: more than 3 min were defined as painful insensitive group (PIS), and less than 1.5 min were defined as painful sensitive group (PS).

During MRI scan, cold bottle test (CBT) was completed. Two fMRI sessions were scanned for each subject. First, the experimenter delivered 26 °C glass bottle to subjects’ right hand. Then, participant received normal-temperature stimulation during fMRI scan. Second, the participant received cold-temperature stimulation through holding 8 °C glass bottle during fMRI scan. Specifically, subjects should rest for at least 3 min after the completion of normal-temperature stimulation scan, and the next scan should be performed when the right hand returns to normal body temperature. It is important to note that all the tasks performed on the MRI scanner required the subject to lie flat, relax and remain still.

Specifically, the acceptability of both CPT and CBT was measured by each subject before the experiment. Subject was free to withdraw from this study. Moreover, all references to 26 °C and 8 °C mentioned in this paper represent the temperature range of 26–30 °C and 8–12 °C, respectively. For the convenience of discussion in this paper, 26 °C and 8 °C are used to represent the above two temperature ranges.

### 2.3. Data Acquisition

All MRI data in the experiment were collected from the MRI Research Center of UESTC and passed through 3T MRI scanner (General Electric Discovery MR750; Milwaukee, WI, USA) with 8-channel head coil acquisition. During scanning, foam pads and earplugs were used to reduce head movement and scan noise, respectively. The main scanning parameters of high-resolution, T1-weighted structural images are as follows: repetition time (TR) = 6.008 ms, echo time (TE) = 1.984 ms, flip angle (FA) = 9°, matrix size is 256 × 256, field of view (FOV) = 256 mm × 256 mm, slice thickness = 1 mm, no gap, 154 slices in total. Gradient–echo planar imaging (EPI) sequence was used to obtain fMRI data. The main scanning parameters were as follows: TR = 2000 ms, TE = 30 ms, FA = 90°, matrix size = 64 × 64, FOV = 240 mm × 240 mm, slice thickness/gap = 4 mm/0.4 mm, a total of 35 slices. Specifically, resting-state sequences were scanned for 250 s over 125 time points.

### 2.4. Data Preprocessing

All functional mappings’ preprocessing steps were performed using the DPABI toolkit (http://rfmri.org/dpabi, accessed on 3 August 2022) and custom scripts in MATLAB. First, T1 structural images were segmented into WM, GM and cerebrospinal fluid (CSF) using DPABI segmentation option and then registered to MNI standard space. Then, the functional images were processed as following: (1) The first 5 volumes (10 s) of data were discarded for magnetization equilibrium. (2) Slice time correction and head motion correction were as follows: the head movement of all subjects was less than 1.5° or 1.5 mm. (3) The noise was removed, and the signals of no interest (including cerebrospinal fluid signals and 24 head-movement parameters) were regressed with a linear regression model [23]. (4) To remove linear drift, regression to the linear trend was performed to correct signal drift. (5) We also excluded participants who have large mean framewise displacement (FD) (e.g., FD > 0.2) as suggested by Jenkinson et al. [24]. The temporal scrubbing was performed by using a linear regression model. (6) Bandpass filtering with a frequency of 0.01–0.15 Hz to reduce the influence of non-neuronal signals on BOLD signals. (7) Spatial smoothness followed. To separate the WM signal from the GM signal, spatial smoothing was performed within the WM mask and the GM mask, respectively. Firstly, the individual space T1 segmentation image of each subject was registered to the functional image of the corresponding subject, which was used to identify the WM and GM mask of the functional image (threshold was 0.5). Functional images were smoothed (Gaussian smooth kernel, FWHM = 4 mm) within GM and WM masks, respectively. (8) Spatial standardization came next: the smoothed image is registered to the standard MNI space and resampled until the voxel size is 3 × 3 × 3 mm^3^. Steps 1–7 are processed on the individual space of each subject to effectively distinguish GM and WM signals at the individual level.

### 2.5. Clustering WM Networks

Firstly, the WM mask at the group level was made according to the segmentation results of T1-weighted images. The steps were as follows: For each subject, we binarized each voxel as WM, GM or CSF based on the probability value of the T1 segmentation images. Then, these masks were averaged among subjects to obtain the percentage of each voxel that was classified as WM or GM. For WM, a voxel with a mean value greater than 60% was defined as a WM voxel, indicating that at least 60% of subjects recognized that voxel as WM. Then, to further remove GM clumps deep in the brain, subcortical areas of the Harvard–Oxford template were removed from the WM mask [25]. Finally, in order to ensure consistency with the functional image, the WM mask was registered in the functional space and resampled to a voxel size of 3 × 3 × 3 mm^3^. The resulting WM mask contained 17,564 voxels. In addition, gray matter mask generation is similar to WM, and we used a relatively loose threshold (20%) to define GM voxel, ending up with a GM mask that contains almost all GM voxels. Finally, we made sure there is no overlap between GM and WM masks.

Secondly, the correlation matrix at the group level was calculated for clustering WM functional networks. The 17,564 voxels of WM mask were resampled to 4353 nodes using the inter-changing grid strategy [26], and the Pearson’s linear correlation coefficient between the time series of each WM voxel. The time series of the sampled node voxels was calculated to obtain the correlation matrix (17,564 × 4353). To obtain an unbiased group level correlation matrix, two averages were performed. Firstly, Matrix-Ⅰ was obtained by averaging the correlation matrices of all subjects in PS group. Then, the correlation matrices of all subjects in PIS group were averaged to obtain Matrix-Ⅱ. Finally, we took the average of Matrix-Ⅰ and Matrix-Ⅱ to get Matrix-Ⅲ. Because the number of people in PS and PIS groups is not equal, if we directly average all subjects, the results may be weighted more heavily for groups with more subjects, resulting in bias.

Finally, the cluster analysis is carried out on WM. In order to obtain different WM functional networks, we use an iterative clustering algorithm called K-means clustering algorithm. Firstly, K-means clustering is performed on matrix Matrix-Ⅲ, and the number of clusters K is traversed from 2 to 22. Previously, excellent researches indicated that clustering according to different features should provide approximately similar results for numbers of clusters [27,28,29]. In order to ensure the stability of clustering results, Matrix-Ⅲ (17,564 × 4353) is randomly divided into four submatrices, namely Matrix-Ⅲ.1, Matrix-Ⅲ.2, Matrix-Ⅲ.3 and Matrix-Ⅲ.4, with the size of each submatrix being 17,564 × 1088. The same cluster analysis was performed for each submatrix separately, and the similarity between cluster results was measured with Dice coefficient. The averaged Dice’s coefficient was used to assess the stability of the number of clusters. Finally, the maximum K value whose Dice coefficient is greater than 0.9 is selected as the final number of clusters.

In view of the reliable and highly consistent results of clustering GM fMRI signals in many studies, this study did not conduct separate clustering analysis of GM voxels. Instead, a GM map in previous literature was selected, which was obtained by using the K-means clustering algorithm mentioned in this study. The results were consistent with the existing brain function map.

### 2.6. FC Analysis of WM Networks

FC matrices were calculated among WM networks, as well as between GM and WM networks as follows: (1) The average time series of all voxels in each cluster are extracted. (2) Pearson’s correlation was performed between two clusters. (3) FC matrix was transformed into Fisher-z score. (4) The group mean FC matrix was obtained.

### 2.7. Statistical Analysis

To assess FC differences between WM and WM networks across groups and stimuli, we performed assessments using repeated measures ANOVA. The significance level was set to *p* < 0.005 (without correction) for statistical analysis. During the post hoc analysis, paired *t*-test was used to assess the difference between two stimulus conditions in each group. Two-sample *t*-test was used to measure the difference between two groups in each condition. To explore the relationship between WM networks and GM networks, we performed similar analysis.

### 2.8. Correlations with Emotion Scales

We used SPSS (SPSS Statistics|IBM) to assess the linear relationship between STAI and the changed FC in two groups, respectively. The age of the subjects was used as a covariate in the regression to reduce the effect of age. The significance level was set to *p* < 0.05 (without correction) for the correlation analysis.

### 2.9. Validation Analysis

To ensure a high reproducibility of our results, the validation analysis was performed through adding the gray matter cluster analysis based on the collected data. Then, the functional connectivity and statistical analyses were performed.

## 3. Results

### 3.1. Demographics and Psychological Characteristics

A subject’s head motion of more than 1.5° or 1.5 mm was excluded. Finally, 26 subjects (mean age 22.96 ± 1.59 years, range 19 to 27 years) in PS group and 27 subjects (mean age 22.48 ± 1.93 years, range 19 to 25 years) in PIS group were obtained. Two groups’ data were all obtained. There were no significant differences in age (*p* = 0.3282), head motion (group: *p* = 0.553; condition: *p* = 0.905; group ∗ condition: *p* = 0.146), SA-score (*p* = 0.3929), TA-score (*p* = 0.5208) between the two groups (Table 1).

### 3.2. WM Networks

Consistent with previous studies, the K-means clustering method can identify symmetric and staggered functional network patterns in WM regions of the brain (Figure 1A). All white voxels were grouped into 2 to 22 clusters, and the Dice coefficient was used to measure the stability of the results. The Figure 1B shows that the maximum K of a Dice coefficient greater than 0.9 is 13, that is, the most stable number of clusters is 13. The resulting 13 WM networks are shown in Figure 1C. See Figure 2 for details.

### 3.3. Alterations in FC between WM Networks

The results of repeated measures ANOVA showed that there was a significant interaction between group and stimulus condition (*p* < 0.005, blue links in Figure 3A). Compared with normal condition, the FC between the WM–frontotemporal network (WM–FTN) and the WM-precentral/postcentral network (WM–PCN) was significantly decreased in the PS group under cold condition. The differences between groups were not statistically significant (Figure 3B).

### 3.4. Alterations in FC between WM and GM Networks

The results of repeated measures ANOVA showed a significant interaction between the group and the temperature stimulus conditions (*p* < 0.005, red links in Figure 3A). Compared with normal condition, the FC between GM–sensorimotor network (GM–SMN) and WM–FTN, GM–posterior cerebellum network (GM–pCBN) and WM–superior temporal network (WM–sTN), GM–pCBN and WM–PCN, GM–anterior cerebellum network (GM–aCBN) and WM–FTN, GM–visual network (GM–VN) and WM–main corpus callosum network (WM–mCCN) were significantly decreased in the PS group in cold condition. Compared with PIS group, the FC between GM–SMN and WM–FTN was significantly enhanced in the PS group under the normal condition. On the contrary, the FC between GM–pCBN and WM–PCN, GM–VN and WM–mCCN was significantly reduced in the PS group under cold conditions, and the other differences were not statistically significant (Figure 3B). Moreover, through validation analysis, these results were also observed through the new gray matter cluster analysis.

### 3.5. Correlation between FC and Scale Scores

The relationship between the FC of WM networks and the scale scores is shown in the Figure 4. We found that in the PS group at normal temperature, the SA-subscale score was positively correlated with pCBN–sTN FC and pCBN–PCN FC, and the TA-subscale score was positively correlated with pCBN–PCN FC. In the PIS group at cold condition, the SA-subscale score was positively correlated with pCBN–PCN FC.

## 4. Discussion

In this study, the correlation analysis of the fMRI data was performed to cluster WM voxels into different WM functional networks, which was consistent with previous findings [29]. Our study has found that the WM functional network is associated with regions involved in the top–down regulation pathways of pain such as the PAG and prefrontal cortex (PFC). In general, this suggests that the relationship between the WM functional network and pain could not be ignored.

It has been found that the cerebellum, which receives descending information from other brain regions and ascending nociceptive information from the spinal cord, is ideally positioned to influence or be influenced by pain processing [30]. Therefore, the cerebellum plays an important role in the descending pain modulatory system. This modulatory system has both promoting and inhibiting effects on pain. For example, previous research found that blocking the action of G protein-coupled receptor 55 (GPR55) in PAG drives the descending pain modulatory system to reduce inflammatory pain [31]. In addition, it has been demonstrated that intense stress and fear are associated with descending inhibition and hypoalgesia, while inflammation and nerve injury are associated with descending facilitation, hyperalgesia and pain [32]. Our study found that FC between GM–pCBN and WM–SMC were significantly reduced in the PS group. Recent findings suggest that the posterior cerebellum may represent a critical modulator in the cognitive appraisal of pain through cortico–cerebellar neural loops, which may have downstream effects on motor function [11,33]. In addition, the sensory network is involved in stimulus localization and intensity coding of pain [34]; the motor network is involved in effective pain avoidance [35]. Therefore, based on the results of this study, we inferred that pCBN might effectively modulate the pain perception by affecting the WM sensorimotor cortical network through the descending pain modulatory system.

It is known that the posterior cerebellum and the fronto-temporal area are the key regions of the default mode network (DMN) [36,37]. Previous studies have found that the PAG is a key region in the pain regulatory pathway, closely coordinated with the DMN, and it is associated with the tendency to mind wander away from pain [38]. Therefore, it is reasonable to suspect that pCBN also performs the same function as PAG in regulating the pain perception. Thus, in this study the changes in FC between the pCBN and the WM–sTN network in this study might indicate that the outcome of individual pain relief was co-regulated by pCBN and DMN on the basis of the regulation of pain perception by PAG.

The cerebellum is known to be involved in higher-order cognitive regulation of the DMN. The DMN, which is involved in self-referential and stream of consciousness processing, is one of the most closely watched networks [39]. Moreover, based on imaging studies of human diseases and EMG studies of animal experiments, altered functional connection within the sTN and the PCN are related to the functional activation of SMC [40,41]. Moreover, the WM–sTN network and the WM–PCN have been confirmed to be associated with SMC [42]. In addition, superior temporal gyrus and pre- and post-central gyrus are important brain regions for sensorimotor information processing. Therefore, our findings indicate that the reduced FC between DMN and sensorimotor networks in the PS group may reflect the need to attenuate the brain’s self-referential connectivity as a means of more effective modulation on external pain stimulation.

Emotion is one of the major changes that accompany the onset of pain. In this study, we used STAI to measure the anxiety characteristics of the subjects. Anxiety is one of the negative emotions. Previous studies have shown that abnormal DMN function may cause negative emotions such as anxiety and depression [43]. It is known that a reduction in negative emotions is significantly associated with a decrease in pain [44]. Pain may be a common symptom and a good indicator of anxiety disorders, while anxiety may also lead to higher levels of pain temporalization [45]. Similarly, Kapoor et al. studied patients with acute pain and showed that the pain intensity was also positively correlated with SA score [46]. This is consistent with our research results. This series of findings provides an effective basis for improving anxiety and pain caused by various diseases.

In this study, subjects were divided into two groups by CPT, corresponding to high and low pain sensitivity, respectively. In addition to the degree of pain sensitivity, the two groups were consistent in age, gender, education and other factors. This allows us to understand the differences in WM activity between insensitive and sensitive individuals. It provides additional reference for improving pain perception. There are also some limitations in this study. First of all, during the WM-cluster analysis, the mean functional connectivity matrix was obtained from all subjects under two temperature conditions. It would miss the specific features of one group under one condition. Thus, the new cluster algorithm should be used to validate and extend our findings. Second, only male subjects were selected in this study. Although many other factors were controlled, the single gender would also limit the experimental results. In future studies, consider studying WM changes in male and female subjects exposed to the same pain stimuli. Finally, during the CBT, the temperature of the ice bottle will change with the time of holding the bottle, the temperature will generate errors, which will also cause errors in the experiment results. In the later study, a water circulation thermostat can be considered for experiments.

## 5. Conclusions

In conclusion, the results of this study suggest that the WM functional network plays a conduction role in pain processing, and changed FC may be related to the descending pain modulatory system. According to our findings, FC reduction between DMN key networks and WM SMC networks is the result of the involvement of pain stimulus localization, intensity coding and pain avoidance. To our knowledge, this study is the first to evaluate FC changes in WM bundles in response to cold pain stimuli in a PS population compared with a PIS population. It highlights the importance of understanding WM function as a component of brain functional networks in pain patients. Our findings provide a new idea around the WM pathway for modern pain research and provide a basis for improving pain experience.

## Figures and Tables

**Figure 1 bioengineering-10-00988-f001:**
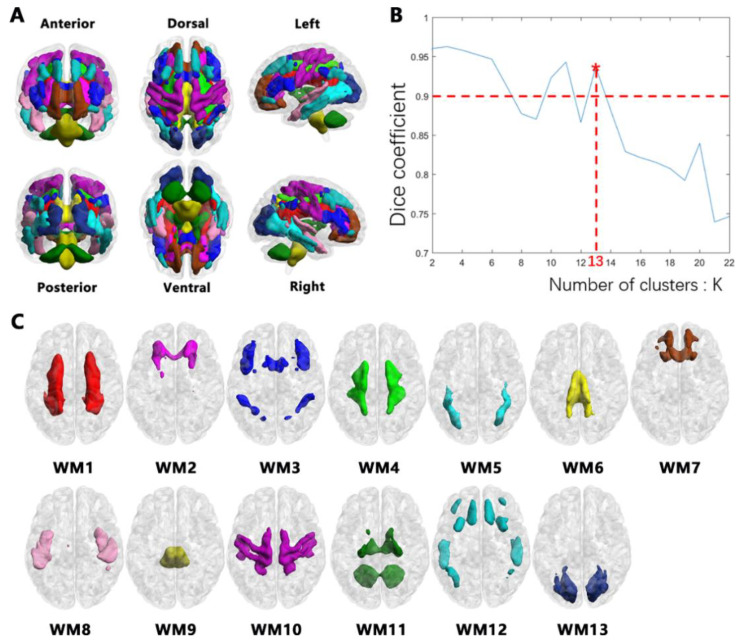
Thirteen WM networks with high stability were obtained using K-means clustering algorithm. (**A**). WM networks in full view. (**B**). Stability of clustering for different numbers of clusters. The maximum K of Dice coefficient greater than 0.9 is 13. “*” denotes the most stable number of clusters. (**C**). Dorsal view of 13 WM networks.

**Figure 2 bioengineering-10-00988-f002:**
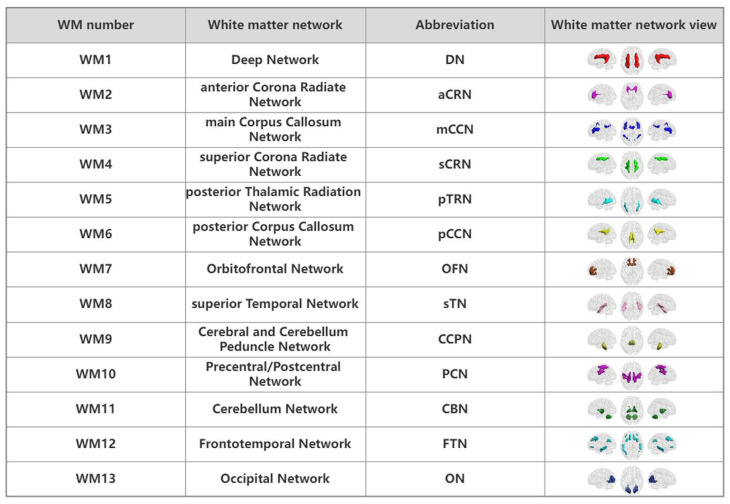
Detail views of the WM functional networks. All WM number, full names, abbreviations and maps have a corresponding relationship.

**Figure 3 bioengineering-10-00988-f003:**
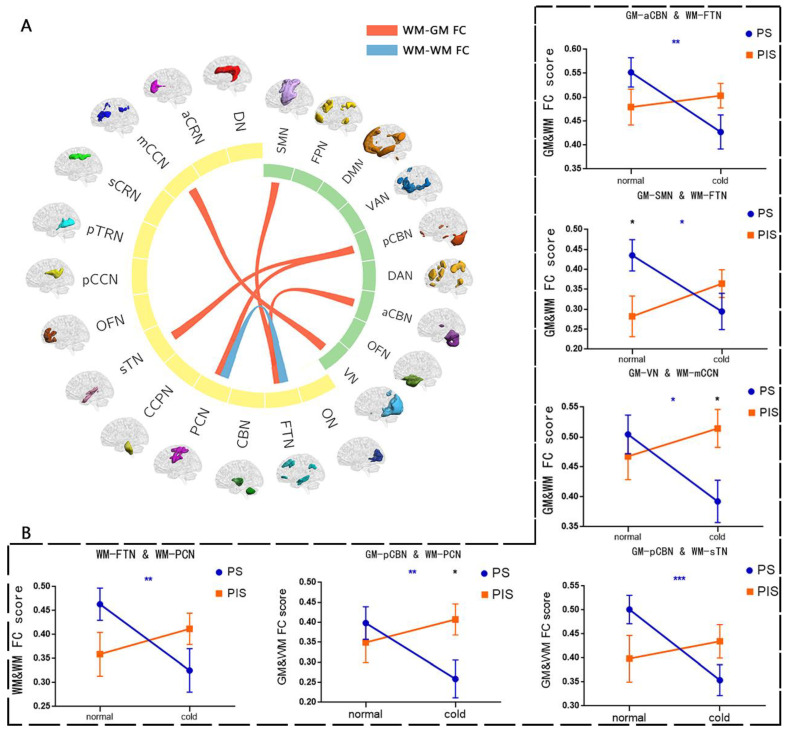
There was a significant interaction between the group and the stimulus condition in the FC between networks. (**A**). The green bands represent the GM network, and the yellow bands represent the WM network. All band results had a *p* value of less than 0.005. (**B**). Results of post hoc tests. Among them, * represents 0.01 < *p* ≤ 0.05, ** represents 0.001 < *p* ≤ 0.01 and *** represents *p* ≤ 0.001.

**Figure 4 bioengineering-10-00988-f004:**
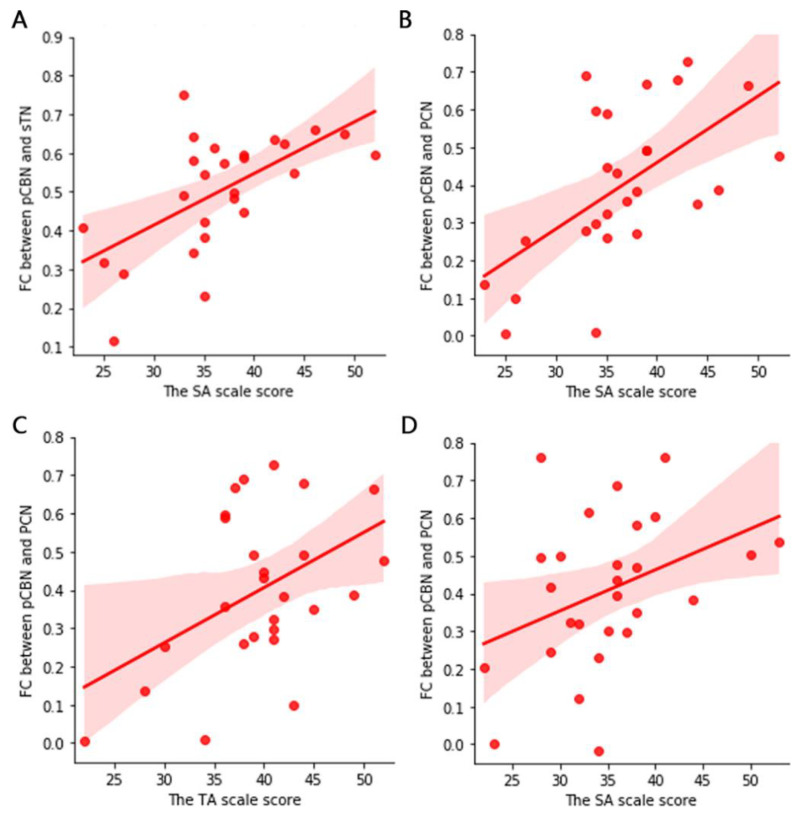
The (**A**–**C**) plot is the correlation results of the PS group under normal condition. (**D**) shows the correlation results of the PIS group under cold condition. In PS group at normal condition, the SA-subscale score was positively correlated with pCBN–sTN FC and pCBN–PCN FC, and the TA-subscale score was positively correlated with pCBN–PCN FC. In PIS group at cold condition, the SA-subscale score was positively correlated with pCBN–PCN FC.

**Table 1 bioengineering-10-00988-t001:** Demographic information and clinical scale scores of the two groups of subjects were collated.

	Pain Sensitive Group (PS)	Pain Insensitive Group (PIS)	*p*-Value
Age (years)	22.96 ± 1.59	22.48 ± 1.93	0.3282 ^a^
Gender (male/female)	26/0	27/0	-
Head Motion (FD)	cold: 0.035 ± 0.016 normal: 0.038 ± 0.015	cold: 0.040 ± 0.018 normal: 0.038 ± 0.011	0.5530 ^b1^ 0.9050 ^b2^ 0.1460 ^b3^
STAI-SA score	36.58 ± 6.81	34.93 ± 6.87	0.3929 ^a^
STAI-TA score	39.50 ± 6.46	38.41 ± 5.59	0.5208 ^a^

Indicated values are shown mean ± standard deviation; FD, framewise displacement; STAI, State-Trait Anxiety Inventory; SA, State Anxiety; TA, Trait Anxiety; a Indicates the *p*-values are from the comparison analysis (two-sample *t*-test). b1, b2 and b3 Indicate the *p*-values are from the comparison analysis (repeated measures ANOVA). b1 Indicates the *p*-value is the group main effect. b2 Indicates the *p*-value is the condition main effect. b3 Indicates the *p*-value is the interaction between group and condition.

## Data Availability

The data presented in this study are available on request from the corresponding author. The data are not publicly available due to their needing to be agreed on by the corresponding author.

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
