# Peer review of "Functional Changes of White Matter Are Related to Human Pain Sensitivity during Sustained Nociception"

_bioengineering, 2023, doi:10.3390/bioengineering10080988_

Round 1

Reviewer 1 Report

Dear Authors,

the research is worth of merit. I enclose a file which mostly contains methodological concerns I would like you may consider.

Good luck, best

Moderate editing of English language required

Author Response

We thank both the reviewers for their thoughtful and insightful comments regarding our study. We have thoroughly revised our manuscript based on their comments.We have added essential information about introduction, methods, result and discussion sections in the main text.

We have addressed each of the comments raised by the reviewers to the best of our abilities.

Reviewer 2 Report

As introduced in the paper, this study is the first one to evaluate functional connectivity changes in the white matter in response to cold pain stimuli in a pain-sensitive population and a pain-insensitive population. This is an interesting point.

Based on the methods introduced in the paper, the cluster method is a process by which voxels in the white matter (or in the grey matter) presenting similar BOLD signal changes form a cluster (a network), assessed by Pearson’s correlation coefficient. Afterward, functional connectivity (FC) between the acquired white-matter networks and gray-matter networks was assessed.

I have several questions and suggestions:

1.     How many fMRI sessions were scanned for each subject? Did all subjects receive high-temperature stimulation first and then low- temperature stimulation (the order of temperature stimuli)?

2.     Why average clustering correlation matrices of the pain-sensitive group and the pain-insensitive group (in Section 2.5: take the average of Matrix-â… and Matrix-â…¡to get Matrix-â…¢)? The two groups experienced different stimuli and therefore may exhibit different white-matter clusters/networks.

3.     Why to divide Matrix-â…¢ (17564×4353) into four submatrices? What is the spatial scope of each submatrix? After dividing, each acquired white-matter cluster/network would be within the scope of a submatrix (i.e., cannot exist in the level of the entire whiter matter but only exist in a 1/4 of the entire spatial scope)?

4.     As to the gray-matter clustering, the stimulation paradigm in the study may not ensure a pattern of gray-matter networks similar to previously detected ones (since the authors introduced cold pain stimuli is novel in the field), and therefore gray-matter clustering based on the collected data may be needed.

5.     In the discussion section, DMN is a focus. However, none of the detected networks present a typical DMN pattern (see Figure 4). Spatial correlation can be calculated between an acquired network and a DMN template, and a high correlation value indicates a similarity.

6.     PAG is another focused point in the discussion section, whereas none of the detected networks contains PAG.

7.     An abbreviation should be given a full name when it appears for the first time, e.g., STAI in line 89, WM-FTN and WM-PCN in line 237, those in lines 244-245, and GM-pCBN and WM-SMC in line 310.

8.     Does the sequence of the 1-13 WM networks have any meaning? For example, the first one has a high value of any index?

9.     In lines 310-311, does “Our study found that FC between GM-pCBN and WM-SMC were significantly reduced in PS group” correspond to “Compared with normal condition, the FC between WM-FTN and WM-PCN was significantly decreased in PS group under cold condition.”(Lines 236-238)?

10.  “Acknowledgments: In this section, you can acknowledge any support given which is not covered by the author contribution or funding sections. This may include administrative and technical support, or donations in kind.”-this should be deleted.

11. In lines 305-307, “we found that blocking the action of G protein-coupled receptor 55 (GPR55) in PAG drives the descending pain modulatory system to reduce inflammatory pain [23]”, whereas [23] is not a study published by the authors.

An abbreviation should be given a full name when it appears for the first time, e.g., STAI in line 89, WM-FTN and WM-PCN in line 237, those in lines 244-245, and GM-pCBN and WM-SMC in line 310.

Author Response

(The authors gave the same response as above.)

Round 2

Reviewer 1 Report

Dear Authors,

thanks for addressing the concerns I had. There are some minor points you probably missed to deal with and they are:

-study design: detail more clearly (is it observational?)

-any statement to guide this study (e.d. STROBE statement)?

-relevant dates: state

-missing values: are there any? how did you manage?

-acceptability of the device and of the test by the subjects: describe.

Thank you, best regards,

 Moderate editing of English language required

Author Response

We apologized for not finishing the reviewers suggestions and questions. We thanked for your thoughtful and insightful comments again. We have revised the related sentences based on your comments.

Best regards

Hui  He

Q1. study design: detail more clearly (is it observational?)

Q2. any statement to guide this study (e.d. STROBE statement)?.

R1-2: Thanks for your nice comments. To clarify these comments, we added the related sentences in line 91-96:”The Strengthening Observational Studies in Epidemiology (STROBE) guideline was followed[22]. Moreover, the study was conducted in accordance with the Declaration of Helsinki, and the protocol was approved by the Ethics Committee of UESTC. To ensure the completeness and accuracy of observational data, a professional master performed experimental procedure. A professional magnetic resonance imaging (MRI) scanner performed data scanning.

      And in line 112-120:”The CPT was produced by immersing subject’s right hand and wrist in a 4-liter tank of water and crushed ice. The water temperature ranged from 0.5 - 1.5ºC. During the subject experienced cold pressor trial, they were requested to place their right hand and wrist in ice water and to leave it there until the pain reached an intolerable level. Subject was not permitted to remain in the ice water for more than 5 minutes. The time was recorded. Subjects were free to withdraw from this study at any time. Then, the painful sensitive and insensitive subjects were grouped based on individuals’CPT tolerance time as follow: more than 3 minutes were defined as painful insensitive group (PIS), and less than 1.5 minutes were defined as painful sensitive group (PS). ”

Q3. relevant dates: state

R3: Thanks for your nice comment. To clarify this comments, we added the sentence in line 99-100:”All subjects involved in this study were recruited from January 2021 to February 2021 at University of Electronic Science and Technology of China (UESTC).

Q4. missing values: are there any? how did you manage?

R4: Thanks for your nice comment. In this study, professional master performed all experimental procedure. There were no missing values in this study. Thus, to clarify this comment, we added sentences in line 93-96:”. To ensure the completeness and accuracy of observational data, a professional master performed experimental procedure. A professional magnetic resonance imaging (MRI) scanner performed data scanning.” And in line 250-251:”Two groups’ data were all obtained.” 

Q5. acceptability of the device and of the test by the subjects: describe.

R5: Thanks for your nice comment. To clarify this comment, we added the related sentences in page 130-131:”Specifically, the acceptability of both CPT and CBT was measured by each subject before the experiment. Subject was free to withdraw from this study.

 Q6. Moderate editing of English language required

R6: We are very sorry for the numerous grammatical errors. To clarify this issue, we have revised these sentences. Furthermore, we have thoroughly checked the manuscript for any grammatical errors.
